# Non-identical moiré twins in bilayer graphene

Everton Arrighi [1,4], Viet-Hung Nguyen [2,4], Mario Di Luca[1], Gaia Maffione[1], Yuanzhuo Hong[1], Liam Farrar [1], Kenji Watanabe [3], Takashi Taniguchi [3], Dominique Mailly [1], Jean-Christophe Charlier [2] & Rebeca Ribeiro-Palau [1]✉

The superlattice obtained by aligning a monolayer graphene and boron nitride (BN) inherits from the hexagonal lattice a sixty degrees periodicity with the layer alignment. It implies that, in principle, the properties of the heterostructure must be identical for 0° and 60° of layer alignment. Here, we demonstrate, using dynamically rotatable van der Waals heterostructures, that the moiré superlattice formed in a bilayer graphene/BN has different electronic properties at 0° and 60° of alignment. Although the existence of these non-identical moiré twins is explained by different relaxation of the atomic structures for each alignment, the origin of the observed valley Hall effect remains to be explained. A simple Berry curvature argument is not sufficient to explain the 120° periodicity of this observation. Our results highlight the complexity of the interplay between mechanical and electronic properties in moiré structures and the importance of taking into account atomic structure relaxation to understand their electronic properties.

When the crystallographic alignment of monolayer graphene and BN is almost perfect (close to zero degrees between layers), the electronic, mechanical and optical properties of graphene are strongly modified[1–4]. This is caused by the combination of two effects: (i) a long-wavelength geometric interference pattern, called a moiré pattern, which effectively acts as a periodic superlattice, and (ii) a local enlargement of the lattice constant of graphene to match the one of BN at the inner part of the moiré pattern, leading to a local commensurate state. Outside of the commensurate areas the accumulated stress, due to the stretching of the lattice constant, is released in the form of out-of-plane corrugations where the stacking order changes rapidly in space[5]. These corrugations have the same periodicity as the moiré pattern. For monolayer graphene, both the long-wavelength pattern and the commensurate state are observed every time one of the layers is rotated by sixty degrees.

The commensurate state creates an imbalance of the interaction that the carbon atoms have with the BN substrate breaking the sublattice symmetry[5]. In monolayer graphene/BN the breaking of inversion symmetry has been proposed as the origin of the opening of an energy gap at the charge neutrality point (CNP)[6] and non-trivial quantum geometry characteristics of the electronic band structure[7]. However, very little is known about how the graphene/BN alignment affects systems with more than one layer, such as bilayer graphene.

Here, we demonstrate that the electronic properties of the commensurate state in a bilayer graphene/BN heterostructure have a hundred and twenty degrees periodicity. We present experimental electron transport measurements in dynamically rotatable van der Waals heterostructures[4] made of Bernal stacked bilayer graphene and BN. Our measurements reveal distinct behaviors for 0° and 60° which we attribute to different electronic band structures generated by different atomic displacements inside the moiré superlattice. However, the observation of the valley Hall effect, only present for 0° of alignment remains to be explained given that the current theoretical model fail to explain this hundred and twenty degrees periodicity.

[1]Université Paris-Saclay, CNRS, Centre de Nanosciences et de Nanotechnologies (C2N), 91120 Palaiseau, France. [2]Institute of Condensed Matter and Nanosciences, Université catholique de Louvain (UCLouvain), 1348 Louvain-la-Neuve, Belgium. [3]National Institute for Materials Science, 1-1 Namiki, Tsukuba, Japan. [4]These authors contributed equally: Everton Arrighi, Viet-Hung Nguyen. ✉e-mail: rebeca.ribeiro@c2n.upsaclay.fr

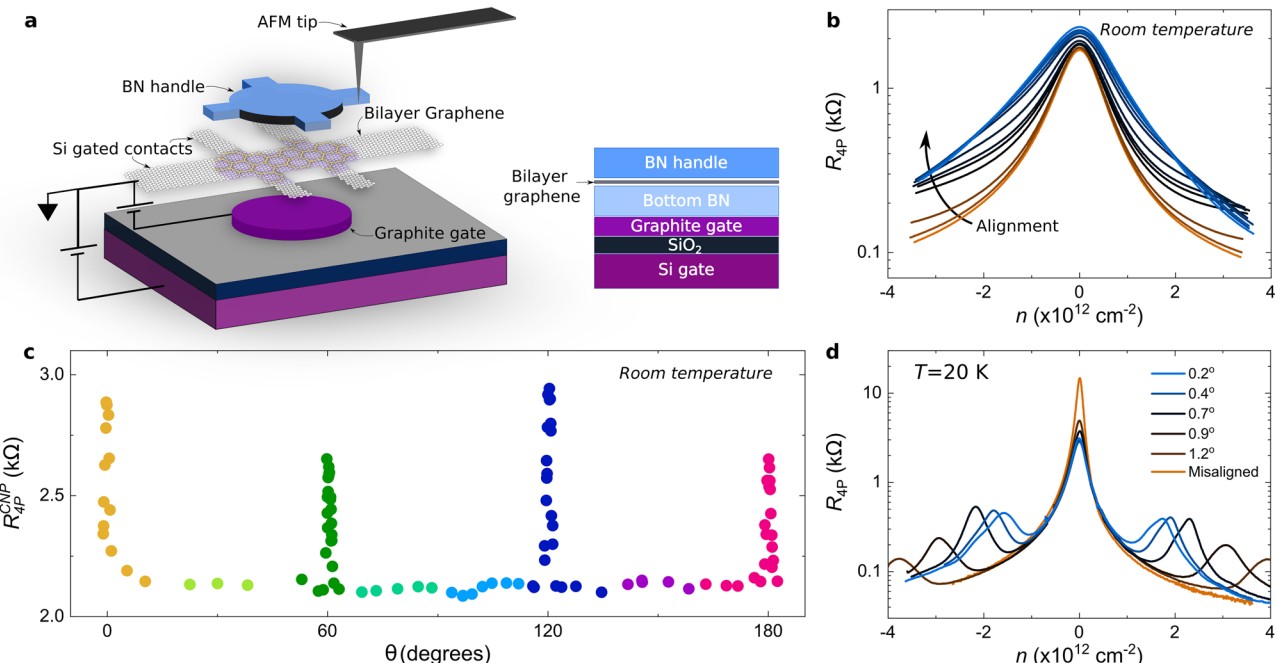

**Fig. 1 | Dynamically rotatable heterostructure. a** Schematic representation (left) of a dynamically rotatable van der Waals heterostructure, the boron nitride between the graphite and graphene layers has been omitted for clarity. The central circular shape on the graphene represents the range of action of the graphite gate and the moiré superlattice. Cross section of the same heterostructure is presented in the right pannel. **b** Four probes resistance measurement as a function of carrier density for several angular alignments of the bilayer graphene and the BN handle, from misaligned (15° - brown curve) to fully aligned (0° - light blue) at room temperature, measurements of sample II. **c** Resistance of the charge neutrality point as a function of the angular alignment, measured with the atomic force microscope, from -2° to 182°, data taken in sample III. **d** Four probes resistance measurement as a function of carrier density for a selection of angular alignments of b at 20 K, sample II. The angular alignment is calculated from the position in energy of the satellite peaks.

## Results

### Angle calibration and room temperature experiments

A schematics of our device and its cross section is shown in Fig. 1a. The dynamically rotatable van der Waals heterostructures are realized as described in[4], with the improvement of having a pre-shaped local graphite gate. The latter controls the carrier density only in the central area of our device, and has the same dimensions as the BN structure used to create the moiré pattern. It is important to mention that the bottom BN and graphene layers are intentionally misaligned, to more than 10°, to avoid the formation of a double moiré[8–10]. The carrier density of the external parts of graphene is tuned by the global Si gate, acting effectively as a tunable contact resistance[11]. The angular alignment of the bilayer graphene/BN heterostructure is controlled in situ by means of a pre-shaped BN handle deposited on top of graphene. This handle can be rotated by applying a lateral force with the tip of an atomic force microscope (AFM), Fig. 1a. The AFM images of the three main positions described in this report: 0°, 30° and 60° can be see in Supplementary Fig. 1. The alignment is fixed at room temperature inside the AFM, using as a reference electron transport measurements (see below), after which the sample is moved to a cryostat for low temperature experiments. The carriers mobility of our samples ranged from 150.000 to 200.000 $cm^2V^{-1}s^{-1}$ for intermediate densities $\pm 0.65 \times 10^{12}$ $cm^{-2}$ at $T < 10$ K. The mean free path was calculated to be between 1.2 μm and 2 μm for the same carrier density and temperature range (see Supplementary Fig. 8 for details). These values of the mean free path are comparable with the nominal dimension of our samples W = 1.7 μm and L = 2.3 μm, reflecting a ballistic transport regime. All measurements presented here were taken using lock-in amplifiers at $f \approx 33.37$ Hz and applied currents of 10 nA. The non-local voltages were measured using a high input impedance voltage amplifiers to ensure the measurement had no leaking current effects. We also ensure that this is not a heating effect by performing the same

measurements at different currents (for more details see Supplementary Fig. 12).

In contrast with monolayer graphene[4], for bilayer aligned with BN the presence of satellite peaks in charge transport measurements - a clear signature of the moiré superlattice - becomes evident only at low temperatures. At room temperature, these satellite peaks are not visible. This is explained by a smaller intensity of these satellite peaks in the bilayer case, which makes them indistinguishable from the CNP at room temperature due to thermal broadening, as can be seen in the full temperature dependence curves of Supplementary Figs. 5 and 6. In the case of the bilayer, the signature of crystallographic alignment is then given by a broadening of the resistance peak around the charge neutrality point (CNP), Fig. 1b. The combination of room and low temperature measurements, Fig. 1b and d respectively, allows us to have a calibration of the angular alignment at room temperature. The broadening of the resistance peak and its corresponding increase in magnitude are observed every sixty degrees of alignment at room temperature, Fig. 1c. However, the maximum of the resistance at the CNP for the aligned position is in fact periodic every hundred and twenty degrees of rotation, Figs. 1c and 2c, with a slow decrease as the moiré length is reduced. Other values, such as the position of the CNP in gate voltage are also affected and hold the same periodicity, see Supplementary Figs. 3 and 4. For reference, and to be consistent among all our samples, we establish that the aligned position with the highest resistance at the CNP will be named 0° of alignment. The features of alignment presented here, such as the observation of the valley Hall effect, are all consistently observed in the alignment with highest resistant at room temperature.

### Local and non-local charge transport response

At low temperatures, for both 0° and 60° of alignment, the local charge transport measurement shows the presence of very well

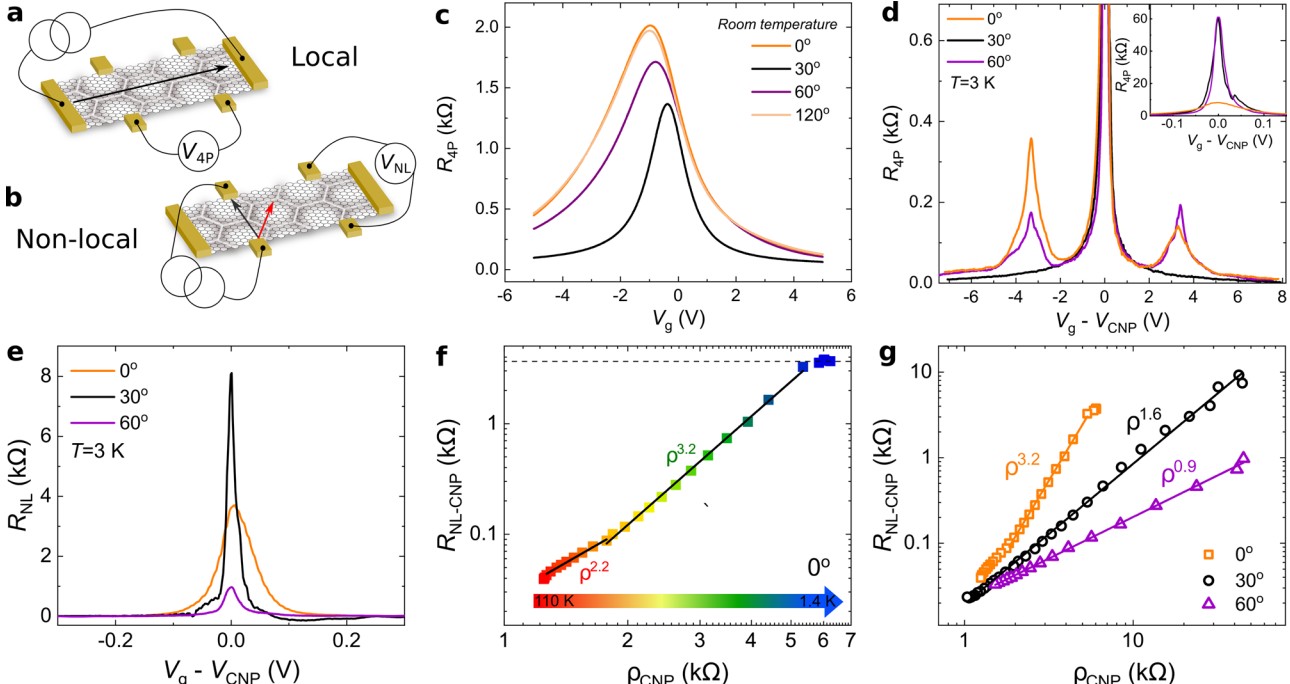

**Fig. 2 | Local and non-local transport measurements. a, b** Artistic representation of the local measurement (**a**) where the four probes local voltage ($V_{4P}$) is measured, and non-local measurement (**b**) where the non-local voltage ($V_{NL}$) is measured. **c, d** Local measurements as a function of the applied gate voltage at room temperature (**c**) and $T = 3$ K (**d**), insert in (**d**): zoom around the CNP. **e** Non-local resistance $R_{NL} = V_{NL}/I$, where $I$ is the applied current, as a function of the gate voltage around the CNP at $T = 3$ K. **f** Non-local resistance as a function of the local resistivity $\rho = (W/L)R_{4P}$, both measurements (local and non-local) are at the CNP, for 0° of alignment, here $W$ and $L$ are the width and length, respectively. The color scale indicates different temperatures. Dashed line indicates the expected value of the non-local resistance for a valley angle of $\pi/2$, see text. Solid black lines are fit to the experimental data using eq. (1), see text. **g** Non-local resistance as a function of the local resistivity, both at the CNP, for different angular alignments. Solid lines are linear fits to the experimental data meant to extract the power law dependence. Both (**f, g**) are measurements between $T = 1.4$ K and $T = 110$ K.

pronounced satellite peaks at both sides of the CNP, Fig. 2d. These satellite peaks are accompanied by a sign reversal of the Hall resistance $R_{xy}$ when a low magnetic field is applied (see Supplementary Fig. 7). In the case of 30°, as expected, these satellite peaks are not present since graphene and BN are completely misaligned and the moiré pattern is absent. From magneto-transport measurements we extract a moiré wavelength of $\lambda = 14.1 \pm 0.4$ nm and $\lambda = 14.5 \pm 0.3$ nm for 0° and 60°, respectively (see Supplementary Fig. 13). The good coincidence between the positions of the satellite peaks and very close values of the extracted moiré wavelength allow us to say that a good alignment is reached in both cases. Notice that at low temperatures the difference between these two alignments can be reduced to a different magnitude of the local resistance at the satellite peak and CNP (see Fig. 2d), which in an experiment using samples with fixed angles would be attributed to a sample-to-sample dependence.

In order to explore more subtle modifications of the properties of this system, we changed the measurement configuration to a non-local one, represented in Fig. 2b. The non-local electrical signal refers to the appearance of a voltage across contacts that are well outside the expected path of the current. This technique is largely used to detect spin/pseudospin signals[12–16]. Non-local signals, attributed to the existence of valley currents, have been measured previously in aligned monolayer graphene/BN[13,17,18], in bilayer graphene aligned with BN[19] and in the presence of a strong displacement field[14,15] as well as in other 2D materials[20]. In this report we focus only on non-local signals at the CNP, since the non-local signals around the satellite peaks are too weak to be studied systematically with our current experimental setup. As in previous reports[13,14], the non-local resistance, $R_{NL}$–Fig. 2e, decays rapidly with carrier density to values lower than our experimental measurement noise. Furthermore, we observe a very strong dependence of the non-local signal with the angular

alignment. Indeed, the maximum value of the non-local resistance at the CNP decreases by a factor four between the measurements at 0° and the one at 60°, Fig. 2e, a contrasting behavior with respect to the local signal where the 0° of alignment has a much smaller signal than the 60°, see Fig. 2d-insert. This suggest that the non-local signal is independent of the local one. In other words that this is not a simple ohmic response.

Plotting the non-local resistance versus local resistivity, $\rho$, at the CNP for different temperatures, for 0° of alignment (Fig. 2f), we observe three regimes: for $T > 40$ K an approximately quadratic dependence $R_{NL} \propto \rho^{2.2}$; for $40$ K $\geq T \geq 12$ K a near-cubic relation, $R_{NL} \propto \rho^{3.2}$, and a saturation regime for <12 K. Let us start the discussion with what happens for $T \leq 40$ K. In analogy with the spin Hall effect[21], this cubic relation is expected in graphene when the valley Hall and inverse valley Hall effects are in operation. In particular, the non-local resistance and local resistivity are related by ref. 22:

$$R_{NL} = \frac{W}{2l_v}(\sigma_{xy}^v)^2 \rho^3 e^{-L/l_v}, \tag{1}$$

when $\sigma_{xy}^v \ll \sigma$. Here, $l_v$ is the inter-valley scattering length, $\sigma_{xy}^v$ is the valley Hall conductivity, $\rho = 1/\sigma$ is the local resistivity and $W$ and $L$ are the width and length of the sample, respectively.

For $T \leq 12$ K, Fig. 2f, there is a clear change of behavior, characterized by a saturation of the non-local resistance. This is consistent with the regime where $\sigma_{xy}^v \gg \sigma$, and the non-local response becomes independent of the local resistivity[22]:

$$R_{NL} = \frac{W}{2l_v}\frac{1}{\sigma_{xy}^v}. \tag{2}$$

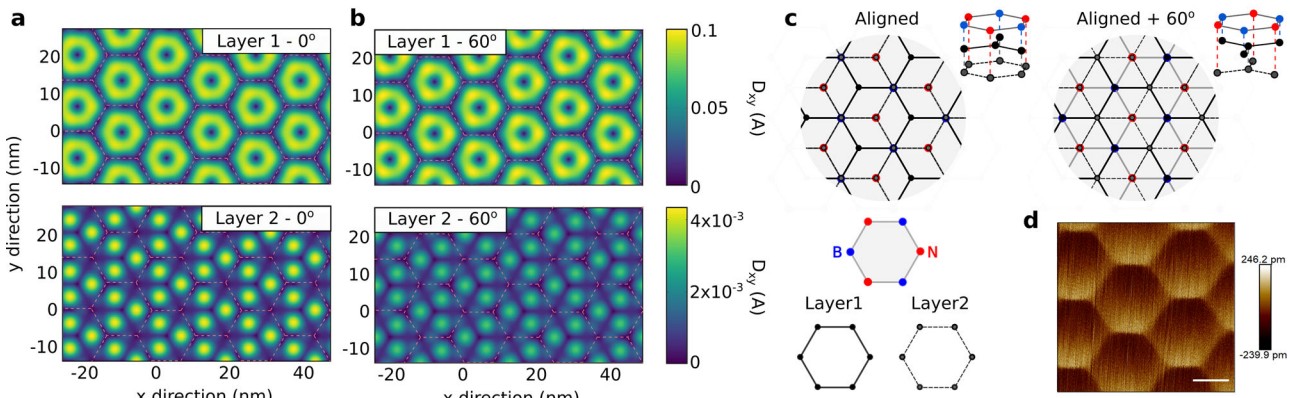

**Fig. 3 | Atomic structure relaxation for bilayer graphene/BN. a, b** Numerical simulations of in-plane displacement, $D_{xy}$, of the carbon atoms for both graphene layers at 0° and 60° of alignment. Pink dashed lines in a and b are guides for the eye to follow the moiré superlattice. **c** Atomistic sketch of a BN/bilayer graphene for 0° (left) and 60° (right) alignments. Description: blue dots boron, red dots nitrogen, black dots carbon atoms of layer one and gray dots carbon atoms of layer two; gray lines represent bonds between boron and nitrogen, solid black line bonds between carbon atoms of layer one and black dashed line represent the bonds between carbon atoms of layer two. **d** AFM image in PeakForce mode, height sensor, of an aligned bilayer graphene BN. Scale bar 7 nm. The angular alignment (0° or 60°) is unknown.

Since the Fermi energy is at the CNP, and the temperature of the system is much lower than the energy gap, the valley Hall conductivity is maximal having a value of $\sigma_{xy}^v = 4e^2/h$ ($e$ is the electron's charge and $h$ is Planck's constant). In this case all occupied states in the valence band contribute to the valley Hall effect[13–15,18,22,23]. Using the saturation value of the non-local resistance, dashed horizontal line of Fig. 2f, and maximum valley Hall conductivity we obtain an inter-valley scattering length $l_v = 1.5$ μm, in good agreement with previous reports[13–15].

This behavior, where the non-local signal is independent of the local response, is characteristic of a system where the valley conductivity is larger than the local conductivity, which implies a fully developed valley Hall effect, where the Hall angle becomes $\pi/2$. For $T > 40$ K we do not have a clear picture of why we observe a nearly quadratic behavior, we attribute this to a mixture of regimes where both the valley Hall effect and ohmic response compete. This regime needs more experimental and theoretical investigation.

If we now change the crystallographic alignment of the layer, by in situ rotation of the BN handle with the AFM tip at room temperature, we can see that the 60° case is very different, Fig. 2g. In this case the relation between the local and non-local resistance is close to linear, $R_{NL} \propto \rho^{0.9}$. This behavior is consistent with an ohmic contribution $R_{NL} = 4\rho e^{-\pi L/W}/\pi$, and can be adjusted by using only the geometry of our sample, as is expected given its van der Pauw geometry. In contrast with the 0° of alignment this linear behavior can be observed over the full temperature range. This striking difference of the development of the valley Hall effect at 0° but absent at 60° reveals that the consequences of the moiré patterns for the two alignments are non-identical. Changing the alignment further to 120° restores the signatures of the valley Hall effect (for more details see Supplementary Figs. 18 and 20).

Interestingly, neither the valley Hall effect, observed at 0°, or the ohmic behavior, observed at 60°, are reproduced in the fully misaligned case, 30°, Fig. 2g. In this case, where no signature of alignment is observed in the local charge transport or magneto-transport measurements, we observe a $R_{NL} \propto \rho^{1.6}$ relation. This behavior, unrelated to the valley Hall effect, could be explained either by the existence of localized and non-topological edge states resulting from edge disorder[24]. Or by the presence of electronic jets separated of ≈60 degrees between them, and consequence of the trigonal warping of the bilayer graphene electronic band structure[25]. The existence of these localized states contrasts with the ohmic response observed for the 60° of alignment. However, in both cases we can hypothesize that the periodic moiré potential may prevent the formation of the localized states (in the same way as artificial disorder generated by a scanning gate does[24]) and that it will modify the trigonal warping of the electronic band structure. Further combinations of scanning gates and electron transport will be necessary to clarify our observation.

## Atomic structure relaxation inside the moiré cell

To understand why these two angular alignments give rise to different behaviors we investigated the in-plane atomic structural relaxation of each layer, Fig. 3a–b. Similarly as the features discussed in[26], the presence of misaligned hBN substrate induces a small crystal field (≈15 meV/nm) in bilayer graphene, leading to a small correction on the simulated bandgap. This correction was added in calculations of Fig. 4a. As illustrated in Fig. 3a and b, the in-plane atomic displacement, $D_{xy}$, clearly shows that, for the layer that is closer to the BN (layer 1), there is an almost circular symmetry around the center of each moiré superlattice (marked by the pink dashed lines). On the other hand, the second layer shows a breaking of this symmetry into a $2\pi/3$ rotational symmetry. The in-plane atomic displacement of the second layer is also at least one order of magnitude smaller than for the first layer. Additionally, we can see that the in-plane atomic displacement on the second layer is larger in the case of 0°. The differences in the stretching of each layer results in the spatial variation of stacking structure of the bilayer graphene (initially, perfect AB stacking) as illustrated in Supplementary Fig. 24.

The difference of the in-plane atomic structure relaxations for 0° and 60° can be traced back to the Bernal stacked configuration, see Fig. 3c. We assume, that at the inner part of the moiré cell the atoms are arranged in a BA stacking, between layer 1 of graphene and the BN layer, here the carbon atoms of layer 1 are preferentially sitting on boron atoms since this is the most energetically favorable configuration[6]. Then the carbon atoms of layer 2 will be sitting on nitrogen atoms. We can see in Fig. 3c that the two stacking configurations, 0° and 60°, turn out to be nonequivalent given that the chemical bonds between them are not arranged in the same way, creating an inhomogeneous stretch of the second layer. This inhomogeneous atomic configuration is at the heart of our non-identical moiré twins. For examples of all the different stacking configurations see Supplementary Fig. 22.

As we mentioned before, the stress generated by the commensurate state is released in the form of corrugations, as in the case of monolayer graphene[5]. These corrugations are transmitted to the second layer and can be observed in the height sensor of our AFM measurements (PeakForce mode) of a different bilayer graphene/BN

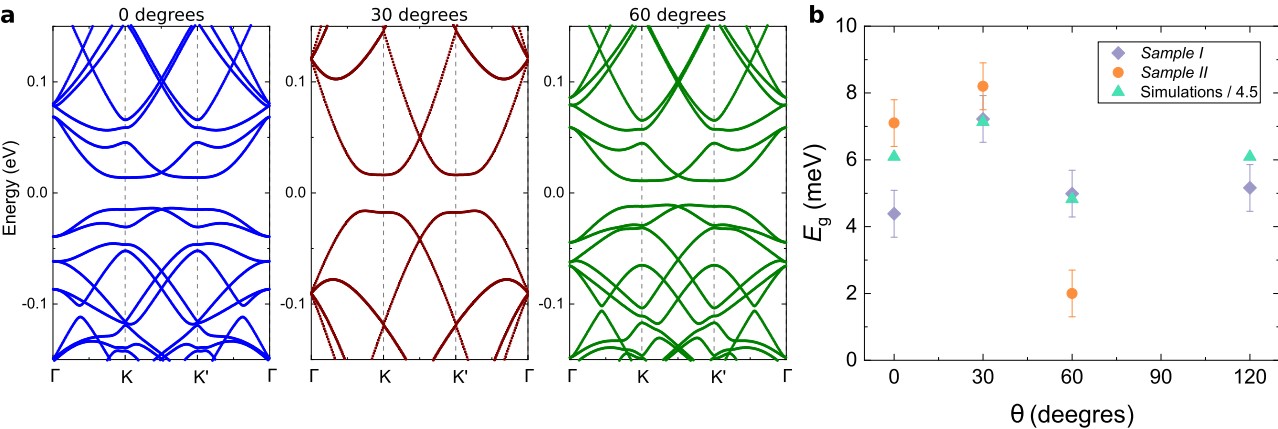

**Fig. 4 | Electronic band structures for different crystallographic alignments.**
**a** Electronic band structures for a relaxed bilayer graphene aligned with BN with 0.1 V/nm of displacement field for 0°, 30° and 60° from left to right. **b** Local energy gaps obtained by thermal activation at different crystallographic alignments (for sample I and II) and theoretical values obtained from (**a**) divided by a factor of 4.5. Error bars represent the standard deviation of our measurements.

aligned sample, Fig. 3d (for more details see Supplementary Fig. 22). In this AFM image, a moiré pattern of $\lambda \sim 14$ nm is clearly observed. Note that the asymmetry on the AFM image and large value on the height of the deformation is an artifact given by the width of the AFM tip being comparable to the size of the features we want to measure (~5 nm). This confirms the existence of a commensurate state and the transmission of the corrugations to the second layer in aligned bilayer graphene/BN heterostructures, supporting our numerical simulations.

The different atomic structure relaxation of the layers results in different electronic band structures for 0° and 60° of alignment (see Supplementary Fig. 25). However, a direct comparison of these with our experimental results is more complicated than it seems since many parameters need to be taken into account, for example the intrinsic displacement field of our samples. Local charge transport measurements show the presence of an energy gap of $E_g^{30°} \approx 7.5 \pm 1.5$ meV at 30° of alignment (for sample I), compared to literature[27] this represents an unintentional displacement field of ~0.1 V/nm. This is not surprising since our devices do not have a top gate to screen external doping deposited on top of the device. Taking into account this unintentional doping and the atomic structure relaxation we calculated the electronic band structures for 0° and 60°, Fig. 4a. These electronic band structures share with our experimental results a small variation of the energy gap with alignment, Fig. 4b, even when the magnitude of the energy gap of the simulation is 4.5 times larger than what we measured in charge transport. Clear differences on the band structure can also be seen in the measurements of non-local resistance as a function of gate voltage and magnetic field (equivalent to magnetic focusing measurements) presented in Supplementary Fig. 21, the explanation of these is out of the scope of this manuscript.

A feature that we do not recover in our experimental measurements is the presence of an energy gap in the valence band for 0° of alignment. Our hypothesis is that this energy gap is too small to be observed in our sample. Even when the quality of our samples is remarkable, compared to previous experiments, the calculated energy gap is about four times smaller than the energy gap at the CNP, which will place it out of reach in our temperature range.

## Discussion

Let's start by discussing the basic charge transport properties of the system. The first sign we present here of the non-identical moiré twins is the difference in the resistance of the CNP at room temperature for 0° and 60° of alignment. Putting these results in the context of the Drude model, and taking into account that we are always working with the same sample, we can attribute this to a different effective mass for

each alignment, coming from distinct electronic band structures, as reflected by our numerical simulations. We will therefore expect to have a heavier mass in the case of 0°, as also suggested by our numerical simulations, see Supplementary Fig. 27. At low temperature the values of the resistance are inverted, now 0° of alignment has a resistance that is about six times lower than the 60° alignment. This can be explain by the presence of the valley Hall effect which will reduce the scattering creating a much better conduction in the 0° case, at low temperatures.

Now we discuss the observation of the valley Hall effect. The most widely used explanation for the existence of the valley Hall effect in aligned graphene/BN is the presence of a Berry curvature[13,18,19,28], detailed in the SI. The Berry curvature has a dependence with the energy gap: it reaches its maximum value for small energy gaps and then it decays rapidly as the energy gap increases[29]. Although this theory explains well the observation of the valley Hall effect in bilayer graphene in the presence of a displacement field[14,15,29], it is in contradiction with our experimental results, where the energy gap amplitude has no incidence in the observation of the valley Hall effect. In Fig. 4 we present two samples which show the valley Hall effect at 0° of alignment. For sample I, the energy gaps do not change significantly among the different alignments, Fig. 4c, and for sample II the smallest energy gap is observed for 60° of alignment. We have also perform numerical simulations of the Berry curvature for the obtained band structures and there are not remarkable difference that could explain our experimental results, see Supplementary Fig. 26.

An alternative explanation to our results could be found in the symmetry of the atomic structure relaxation patterns, Fig. 3a and b. The atomic structure relaxation of the second layer creates a $2\pi/3$ pattern and therefore an anisotropic strain. The particular way in which this strain is applied has been predicted to create a strong gauge field that effectively acts as a uniform magnetic field[30]. This gauge field vector potential has opposite signs for each valley, making possible to have a valley separation and therefore a fully developed valley Hall effect. However, using transport measurements we do not have access to the values of pseudo-magnetic field. We observe a shift in gate voltage of the CNP when the devices are aligned, see Supplementary Figs. 3 and 4. This shift is generated by the change in the work function of graphene induced by strain[31,32]. Unfortunately, these measurements can only be taken as a signal of a larger strain but cannot be used to calculate the pseudo-magnetic field of the system given that they represent an average over the whole device. To prove this theory, local measurements such as scanning tunneling microscopy, will be required.

Spatially varying regions of broken sublattice symmetry: recent theoretical calculations propose that the valley Hall effect observed in monolayer graphene aligned with BN[13,17,18] originates from the spatial variation of the broken sublattice symmetry[33]. If this effect is at the origin of the valley Hall effect in monolayer graphene the picture becomes more complicated when dealing with bilayer graphene. Following the results of our numerical simulations we can say that the spatial variations of broken sublattice symmetry will be different between the two layers, and it will always exist for the first layer. It is then not evident why the valley effect is observed for only one of the two layer alignments, and clearly, further numerical investigations would be needed to clarify the situation.

To conclude, our experimental results show the existence of non-identical moirés in bilayer graphene aligned with BN. We attribute this difference to the atomic structure relaxation of the commensurate state, which modifies the band structure of bilayer graphene in different ways for 0° and 60° of alignment. The observation of the valley Hall effect with a hundred and twenty degrees periodicity cannot be explained by current theoretical model. We hope that our experimental results inspire further theoretical and experimental developments to address the existence of the valley Hall effect in this system.

## Methods

### Sample fabrication

All samples were fabricated by dry transfer of 2D materials, following the technique explained in ref. 4. Pre-shaped bottom gates were fabricated by e-beam lithography and oxygen plasma etching, then a stack of BN/Graphene was deposited on top of the gates using the flip stack technique. This BN/Graphene stack has been intentionally misaligned to ≈10°, to avoid double moiré effects. Angular alignment of the BN handle and graphene structure is achieved using an AFM in contact mode to push the BN handle structure, following the technique explained in ref. 4.

### Local and non-local measurements

Low temperature transport measurements were performed at 10 nA using lock-in amplifiers and low frequency. In the case of non-local measurements a high impedance amplifier was also implemented to avoid any leaking currents. Also for the non-local measurements, we use an OPA to keep the potential of the sample constant.

### Numerical simulations

The atomic-structure relaxation of h-BN/graphene layers is obtained using molecular dynamics with classical potentials. In particular, intra-layer forces are computed using the optimized Tersoff and Brenner potentials[34], while inter-layer interactions are modeled using the Kolmogorov-Crespi potentials[35]. In order to model the moiré structure created when stacking the BN handle (multi-layer h-BN) on graphene, our simulations consider a simplified atomic system consisting of a bilayer graphene coupled to a flat monolayer h-BN. The lattice mismatch (-1.8%) between BN and graphene is taken into account. The atomic structure is optimized until all force components are <1 meV/atom. After relaxation, the electronic structure is computed by constructing and solving the corresponding $p_z$ tight-binding Hamiltonian[36]. The large tight-binding Hamiltonian matrix of moiré structures is diagonalized using conventional eigenvalue calculations of large-sparse matrices.

## Data availability

The Source Data underlying the figures of this study are available at https://doi.org/10.5281/zenodo.7982000. All raw data generated during the current study are available from the corresponding authors upon request.

## Code availability

Codes used to support our findings are available upon request to Viet-Hung Nguyen and Jean-Christophe Charlier.

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

## Acknowledgements

The authors acknowledge discussions with Ulf Gennser, Marco Polini, Herve Aubin, J.I.A. Li and Justin Song. V.-H.N. thanks Dr. Xuan-Hoang TRINH for his helps in implementation of numerical codes to compute the lattice atomic structure relaxation. This work was done within the C2N micro nanotechnologies platforms and partly supported by the RENATECH network, the General Council of Essonne and the DIM-SIRTEQ. This work was supported by: ERC starting grant N° 853282 - TWISTRONICS (R.R-P.), Fédération Wallonie-Bruxelles through the ARC Grant N° 21/26-116 (V.-H.N. and J.-C.C.), Graphene Flagship Core3 N° 881603 (V.-H.N. and J.-C.C.), Flag-Era JTC projects "TATTOOS" N° R.8010.19 (V.-H.N. and J.-C.C. and R.R.-P.) and "MINERVA" N° R.8006.21 (V.-H.N. and J.-C.C.), Pathfinder project "FLATS" N° 101099139 (V.-H.N. and J.-C.C. and R. R.-P.), Fédération Wallonie-Bruxelles through the ARC Grant N° 21/26-116 (V.-H.N. and J.-C.C.) and EOS project "CONNECT" N° 40007563 (V.-H.N. and J.-C.C.), and from the Belgium F.R.S.-FNRS through the research project N° T.029.22F (V.-H.N. and J.-C.C.).

Computational resources have been provided by the CISM supercomputing facilities of UCLouvain and the CE CI consortium funded by F.R.S.-FNRS of Belgium N° 2.5020.11 (V.-H.N. and J.-C.C.).

## Author contributions

R.R.-P. and D.M. designed the experiment. E.A., M.D.L., Y.H. and L.F. fabricated the devices for electron transport measurements. G.M. and M.DL. fabricated the samples for structural characterization and performed the AFM measurements. E.A., M.D.L., Y.H., L.F. and R.R-P performed the electron transport experiments and analyzed the data. T.T. and K.W. grew the crystals of hexagonal boron nitride. V.-H.N. and J.-C.C. performed the numerical simulations and participated to the data analysis. All authors participated to writing the paper. E.A. and V.-H.N. contributed equally to this work.

## Competing interests

The authors declare no competing interests.
