## [Peer Review File · Nature Communications]

Non-identical moire twins in bilayer grapheneREVIEWER COMMENTS

Reviewer #1 (Remarks to the Author):

The manuscript entitled Non-identical moiré twins in bilayer graphene from E. Arrighi et al reported different nonlocal transport behaviors for bilayer graphene aligned with BN at twisted angles of 0 and 60: a fully developed valley Hall is present at 0 angle, while absent at 60. No obvious differences are observed in the local transport measurement. The difference in the nonlocal measurement is contributed to the general argument of symmetry breaking between 0 and 60 alignment between bilayer graphene and BN, and the authors give numerical calculated results of the relaxation of the atomic structures. At last, the authors challenge the current understanding of the valley Hall effect in bilayer graphene on BN originating from Berry curvatures.

The experimental results of the nonlocal transport are surprising because the nonlocal signals are believed to originate from the valley Hall effect and non-zero Berry curvatures in graphene. The authors have a unique technique to rotate the top BN within one device, which in principle can well study the different transport behaviors due to different alignments between graphene and BN. I do, however, have several questions that should be addressed before I can recommend publication.

(1). The advantage of the dynamically rotatable van der Waals heterostructures devices shown in this manuscript is the ability to change the twisted angle between bilayer graphene and top BN within one sample, which can, in principle, avoid other aspects due to different sample qualities. Since the nonlocal data at 0 and 60 are so different, and the authors are trying to challenge the understanding of Berry curvature induced valley Hall effect, the result when rotating back to 0 or at 120 should be produced. In the SI, the authors have shown such a figure at 120 in Fig. S7, but one of the contacts was broken, which result in a non-ideal result by a different measurement geometry. The authors also show the nonlocal data in a non-optimized sample in Fig.S8, which they claim to reproduce their main data in the main text. I think either the data at 120 or the non-optimized device data can claim reproducibility at a publishing level. I understand the sample and the rotating is not easy, but the authors should take efforts to reproduce the data in a second device at 0, 60, and 120 (or back to 0) degrees.

(2). The manuscript tries to connect the nonlocal signal (or valley Hall) with the non-identity in 0° and 60° alignments between bilayer graphene and BN. I try to understand the logic as below: the nonlocal measurement find different behaviors between 0 and 60 angles, and there is a difference in the atomic configuration/commensurate states between the moiré twins, so the authors think the differences in the nonlocal measurement come from the different atomic configurations. I think in general, it might be correct since the 120° symmetry does exist in bilayer graphene/BN moiré. But the argument is weak. There are no necessary connections between atomic configurations and the valley Hall effect. To explain the difference in the nonlocal measurement, there are some possible trivial explanations:

1. It is possible that mechanical rotating generates local strains in bilayer graphene, and therefore changes the local Berry curvatures in part of the sample, leading to valley Hall signals. Since the induced strain pattern can be very different after each rotation, the valley Hall signals are different.
2. Berry curvatures induced valley Hall in bilayer graphene should be different at different out-of-plane

electric fields. Is it possible that the rotation can induce an electric field or local electric field? For example, there exist charged defects in the top and bottom BN, and therefore have a local electric field pattern. The pattern will change after rotating the top BN and therefore influence the valley Hall current.

3. There exist AB/BA domain walls in bilayer graphene, it is possible that the domain walls change the geometry, are generated, or are killed by rotating the sample.

I am sure there are more possible scenarios to explain the nonlocal signal differences for the moiré twins. I am not saying the scenario in this manuscript must be wrong, but the authors just bring out a very general symmetry argument and atomic relaxation calculation to support the symmetry argument, which is not related to the valley Hall effect at all.

I have some minor questions and comments:

(3) For the local measurement at low temperature (in Fig.2b), two resistance peaks at CNP are almost indistinguishable for 0° and 60° . (It is, in fact, hard for readers to identify since data are plotted in thick lines, hidden by each other.) However, from the demonstrated band structure with relaxation in this manuscript (Fig.S11), the gap near zero energy in 60° is about twice larger than that of 0° . Do the authors see this difference in the experiment? From Fig. S5, the slopes of 0 and 60 degrees look similar. How to understand that?

(4) From the calculations, there is a gap at the satellite peak in the hole side. In the SI, the authors claim they didn't see a thermally activated gap in local transport measurement because the gap is small and "is washed out by disorder". But at the same time, the authors claim that the energy gaps of local and nonlocal at CNP "are at least are a factor of two larger than previously reported, this reaffirms the high quality of our samples". So can the authors estimate the disorder level? How does this disorder level compared with the devices in the cited references(the references in SI are missing, so I don't know which reference should I refer), and is it reasonable for this disorders wash out the gap of 5-10 meV at satellite peaks (I get this gap size from the calculated results in Fig. S11).

(5) Since the nonlocal measurement is done at the insulating state, so the current pathway would be very complicated. I think the authors should be very careful. As the authors mentioned, they use "an operational amplifier to keep the voltage of our sample balanced and high impedance amplifiers to avoid any current leak". I believe the authors have done the test, but I encourage them to show the test results in the SI. For example, does the nonlocal resistance depends on the excitation current? Does the nonlocal signal agree with Onsager's relation?

(6) The authors mentioned that the localized edge states resulting from disorder give rise to the nonlocal signal at 30 degree, and the edge states disappear at 60 degree. That is very confusing that the edge states or the disorder can be different at different alignments with BN. Can the authors explain more about this? And a good experiment to test can be an edge length-dependent nonlocal measurement.

(7) How can misaligned bilayer graphene on BN generate a gap, which is even larger than the aligned case? (FIG.2c). Why the non-local signal exhibited at 30° is strongest? (FIG.2e).

(8). The authors should pay attention to the paper writing:

- (a). Please check the format of references in the supplementary thoroughly and eliminate those strange “[?]” typos.
- (b). In Fig.S5a and S5b, data are displayed on a logarithmic scale in very strange numerical labels, which are difficult for readers.
- (c). As mentioned before, the curves in Fig.2b are too thick. This way of presenting data makes it difficult for readers.
- (d). In Fig. S8, the y-axis has two ‘0.1’ labels.
- (e). In Fig.S5, it would be more comparable for the data if the y-axis can be unified into one physic quantity, but not σ for the local measurements and G for the non-local measurements. And please take care of the typos in the figure caption.

In summary, I cannot recommend a publication in Nature Communications at this stage.

Reviewer #2 (Remarks to the Author):

Dear Editor,

Arrighi et al. reported on local and non-local electronic transport data obtained from the hBN/bilayer graphene/hBN Hall bar device, where the top hBN layer is controllably aligned at different angles with the underlying graphene. In such systems, the crystallographic alignment of two crystals with dissimilar lattice constant produces a periodic moiré potential, which ultimately leads to the valley Hall effect associated with the gap opening and finite Berry curvature. In their study, the authors went one step forward and probed the $\pi/3$ rotational periodicity of the valley Hall effect assumed from the crystal symmetry. To do that, the relationship between the local and non-local signal at four different misalignment angles, 0, 30, 60 and 120 degrees, has been compared. Unexpectedly, at 60-degree misalignment, the non-local vs local signal dependence demonstrated the behaviour typical for ohmic transport (linear dependence) instead of the anticipated valley Hall effect observed for the complete alignment (cubic dependence). The valley Hall effect was claimed to recover at 120 degrees. The authors attributed the absence of the valley Hall effect-like response at 60 degrees to the spatial inhomogeneity of the lattice strained in the moiré commensurate state. The potential presence of the inhomogeneity was illustrated by molecular dynamics simulation of the atomic relaxation of bilayer graphene/hBN moiré superlattice. However, the detailed explanation of how the existence of the commensurate state challenges the current explanation of the valley Hall effect and its relation to the Berry curvature and band gap is absent.

Another observed feature is the quadratic dependence of the non-local vs local signal for the fully misaligned case (30 degrees), untypical for ohmic behaviour expected for a fully suppressed valley Hall effect at this rotation angle. The authors attribute this to the possible co-existence of the disorder-induced non-topological edge states and bulk conduction but do not present a more quantitative explanation.

Although the results reported in this paper are very interesting and must be conveyed to the scientific community, I do not recommend it for publication in its present form. The paper requires significant improvement in terms of the data quality and consistency of the suggested explanation. To proceed with

the improvement, I would make the following suggestions on the content and interpretation of the reported data:

- 1) More RNL vs ρ data taken at different misalignment angles is needed. If the authors are correct in their interpretation based on the inequivalent atomic displacement for 0 and 60-degree rotation, then they should confirm the $2\pi/3$ periodicity for all three distinct dependencies of the non-local signal. Namely, data presented in Fig3b should be repeated for 0, 30, 60, 90, 120, 180, 210, 240, 270, 300 and 330 degrees. Note that the data shown in Fig.S7 does not clearly show that the cubic dependence of the non-local signal recovers at 120 degrees of rotation; thus, more data is needed to prove that.
- 2) The trend of the RNL vs ρ data taken at 30 degrees should be explained self-consistently, considering the amount of disorder present in the sample. The authors emphasised the importance of disorder but did not attempt to quantify and compare it between the two samples. It should be assessed either through charge carrier mobility or the width of the Dirac peak at low temperature and correlated with the observed trends of the non-local signal.
- 3) A control experiment with no top hBN handle should be performed to understand the influence of the bottom hBN layer and eliminate the effect of the unintentional alignment of graphene with the bottom hBN. For consistency, the same samples should be used with the top handle dragged away from the active region of the Hall bar.
- 4) The suggested explanation should be re-evaluated considering the above experimental data.

Sincerely,

Reviewer #3 (Remarks to the Author):

The paper by Ribeiro-Palau and co-workers exploits a system with dynamically tunable twist angle to study bilayer graphene rotated with respect to h-BN. This technique was pioneered by her when she was at Columbia and is an important new experimental probe for these twisted 2D materials. The advantage of this technique is that it keeps all the sample properties unchanged isolating only the effect of the twist angle.

The present work compares bilayer graphene twisted at 0, 30 and 60 degrees relative to the top h-BN handle that can be changed in situ. These results as a methods paper would be interesting in and of itself even if the authors did not uncover any new physics.

With the new physics they find, the paper is potentially suitable for Nature Communications. The authors need to address some technical concerns to ensure that the conclusions are robust. But I am optimistic that the authors will be able to do this.

The new physics involves finding a valley-hall effect appearing only for zero twist angle and not for either 30 degrees or 60 degrees. This is despite the three states having roughly the same energy gaps (within experimental error bars). The authors attribute this difference to differences in lattice relaxation, which is a plausible explanation. Without relaxation, the 0 degree and 60 degree states are "twins"; hence the

title of the paper.

A: Experimental Claims

(1) At a very basic level, the non-local measurements are done at the charge neutrality point where disorder effects are strongest. It is not uncommon in this field to mark off a window close to CNP where disorder effects are dominant, and the behaviour is different from elsewhere. Can the authors be sure that their measurements are outside this “disorder window”?

(2) While the local transport on the scale of Fig. 2b show identical behaviour for zero and sixty degrees, a closer look show that the local resistivity is very different, eg. in Fig. S5, the low temperature saturation at 5K is very different between the red and blue symbols. So is it really correct statement to say that this difference in relaxation only shows up in the non-local signal and not the local signal?

I agree that the 30-degree sample does not show a valley Hall effect (e.g. the black curve in Fig. 3b exceeds the theoretical maximum shown as a dashed line in Fig. 3a). And I accept that the authors have put forward only a speculative explanation for the $\rho^{1.6}$ scaling since this is not central to the main claims of the paper. However, the comparison with the 60-degree sample should be considered a bit more carefully. The main claim of this paper rests almost entirely on purple curve having $\sim\rho^1$ over the entire temperature range and the orange curve having three distinct regimes ($\sim\rho^2$, $\sim\rho^3$, and $\sim\rho^0$). Here are some questions about this claim:

(3) Is the temperature range for the 3 curves in Fig. 3b the same?

(4) The authors write that the purple curve “can be adjusted by using only the geometry of our sample”. Was this experiment done with a different sample geometry? This would be a convincing argument;

(5) Can the authors plot the 60-degree data also in Fig. 3a so that it can be compared with the zero-degree data on the same scale? In the larger scale of Fig. 3b, only the $\sim\rho^3$ regime can be clearly seen.

B: Theoretical Claims

The relaxation model is nice, but I hope the authors can do a bit more.

(1) What is it about the relaxation that keeps the local gaps the same (within error bars) between 0 and 60 but changes the non-local gap by more than a factor of 2.

(2) The authors should make a qualitative argument why either having relatively more circular symmetry in the first layer or a larger atomic relaxation in the second layer for 0 degrees is more favourable for the valley Hall effect.

(3) Similarly, the authors should make a qualitative argument for what happens to the Berry curvature when there is relatively more circular symmetry in the first layer or a larger atomic relaxation in the

second layer.

C: Minor questions...

(1) Why are the satellite peaks washed away at room temperature? Can a simple thermal broadening model capture this effect, or is there something more going on?

(2) Most references in the supplemental materials appeared as [?]

Point-by-point response to the reviewers' comments Nat Comms

We thank the three reviewers for the in depth review of our manuscript. This has been a great opportunity and allowed us to have a different point of view about our results. In particular, we realized that our claims were not clearly expressed and we have made an effort to clarify our message giving a stronger emphasis to the differences of the electronic band structure of each fully aligned system reflected not just in charge transport at low temperatures but also at room temperature.

REVIEWER #1 (REMARKS TO THE AUTHOR):

The manuscript entitled Non-identical moiré twins in bilayer graphene from E. Arrighi *et al.*, reported different nonlocal transport behaviors for bilayer graphene aligned with BN at twisted angles of 0 and 60: a fully developed valley Hall is present at 0 angle, while absent at 60. No obvious differences are observed in the local transport measurement. The difference in the nonlocal measurement is contributed to the general argument of symmetry breaking between 0 and 60 alignment between bilayer graphene and BN, and the authors give numerical calculated results of the relaxation of the atomic structures. At last, the authors challenge the current understanding of the valley Hall effect in bilayer graphene on BN originating from Berry curvatures.

Response: The reviewer is right we challenge the current understanding of the origin of the valley Hall effect in bilayer graphene aligned with BN as a solely consequence of the changes in the energy gap of the electronic band structure, which implies a change in the Berry curvature. Instead we propose that this phenomena is more rich and is strongly tied to the atomic relaxation inside the moiré supercell, in the commensurated state. However, so far we have not found a good theory that helps to explain the existence of the valley Hall effect for only one alignment.

The experimental results of the nonlocal transport are surprising because the nonlocal signals are believed to originate from the valley Hall effect and non-zero Berry curvatures in graphene. The authors have a unique technique to rotate the top BN within one device, which in principle can well study the different transport behaviors due to different alignments between graphene and BN. I do, however, have several questions that should be addressed before I can recommend publication.

(1). The advantage of the dynamically rotatable van der Waals heterostructures devices shown in this manuscript is the ability to change the twisted angle between bilayer graphene and top BN within one sample, which can, in principle, avoid other aspects due to different sample qualities. Since the nonlocal data at 0 and 60 are so different, and the authors are trying to challenge the understanding of Berry curvature induced valley Hall effect, the result when rotating back to 0 or at 120 should be produced. In the SI, the authors have shown such a figure at 120 in Fig. S7, but one of the contacts was broken, which result in a non-ideal result by a different measurement geometry. The authors also show the nonlocal data in a non-optimized sample in Fig.S8, which they claim to reproduce their main data in the main text. I think either the data at 120 or the non-optimized device data can claim reproducibility at a publishing level. I understand the sample and the rotating is not easy, but the authors should take efforts to reproduce the data in a second device at 0, 60, and 120 (or back to 0) degrees.

Response: We understand the concerns of the reviewer with respect to the reproducibility of our data. As shown in the supplementary material, we have produced new samples. However, for reasons that are out of our control our cryogenic system has been under repairs intermittently for much longer than we expected. Allowing us to have only incomplete sets of low temperature data. For this reason we have decided to also include room temperature experiments. In this concern we added to the main text Figure 1c, Figure 2c and Figures S4 and S5, which attest for a hundred and twenty degree periodicity of the charge transport properties, at least at room temperatures, for the main sample of the manuscript (sample I), the non-optimized sample (sample II) and a new sample (sample III). We understand this is not what the reviewer wanted to see but we have done (and we are still doing) everything in our power to get our cryogenic system back to work. We have also isolated the non-local resistance versus local resistivity curve at 120° curve to better show its nearly cubic dependence and as we explain in the text the change in the saturation value is related to the change in geometry.

(2). The manuscript tries to connect the nonlocal signal (or valley Hall) with the non-identity in 0° and 60° alignments between bilayer graphene and BN. I try to understand the logic as below: the nonlocal measurement find different behaviors between 0 and 60 angles, and there is a difference in the atomic configuration/commensurate

states between the moiré twins, so the authors think the differences in the nonlocal measurement come from the different atomic configurations. I think in general, it might be correct since the 120° symmetry does exist in bilayer graphene/BN moiré. But the argument is weak. There are no necessary connections between atomic configurations and the valley Hall effect.

Response: The connection we are trying to establish, or the logic in our results, connect the non-identical moiré twins with the different band structures generated by different atomic relaxations inside the moiré cell. Although we find that the band structures are different we cannot find a theory that explains the existence of the valley Hall effect in only one of the aligned positions.

To explain the difference in the nonlocal measurement, there are some possible trivial explanations: 1. It is possible that mechanical rotating generates local strains in bilayer graphene, and therefore changes the local Berry curvatures in part of the sample, leading to valley Hall signals. Since the induced strain pattern can be very different after each rotation, the valley Hall signals are different.

Response: The strain the reviewer is referring to it is not the same strain as we observe inside the moiré supercell, originated by the atomic reconstruction effects in large moiré (i.e., aligned) systems. If we understand correctly the strain the reviewer is making reference to is one that is not necessarily generated by the moiré superlattice. This scenario is possible but unlikely to generate a hundred and twenty degrees periodicity and also won't be the same across different samples. We have demonstrated in our manuscript that the signatures of a strain (or atomic relaxation) are reproduced every hundred and twenty degrees and across different samples.

2. Berry curvatures induced valley Hall in bilayer graphene should be different at different out-of-plane electric fields. Is it possible that the rotation can induce an electric field or local electric field? For example, there exist charged defects in the top and bottom BN, and therefore have a local electric field pattern. The pattern will change after rotating the top BN and therefore influence the valley Hall current.

Response: The energy gap of bilayer graphene can be controlled by out-of-plane electric field and the energy gap is a very important parameter for the Berry curvature and therefore to the presence of Berry curvature induced valley Hall effect, as it has been already established for Bernal stacked bilayer. However, the scenario where the rotation can induce an electric field with a given periodicity is very unlikely. As an example here we present results over different devices which show the same hundred and twenty degrees periodicity.

3. There exist AB/BA domain walls in bilayer graphene, it is possible that the domain walls change the geometry, are generated, or are killed by rotating the sample.

Response: The AB/BA domain walls in bilayer graphene (for example the ones studied in PNAS 110, 11256 (2013)) are indeed a strong point. However, these will need to be generated/killed every 120° , which seems unlikely. Additionally, this will mean that in every sample we have made we encounter such a domain walls in the very small area of our samples. On a different but related note, we computed the possible existence of AB/BA regions created by the moiré potential in the supplementary material. Unfortunately, at this moment we do not have enough information about these regions to point them out as the origin of the valley Hall effect.

I am sure there are more possible scenarios to explain the nonlocal signal differences for the moiré twins. I am not saying the scenario in this manuscript must be wrong, but the authors just bring out a very general symmetry argument and atomic relaxation calculation to support the symmetry argument, which is not related to the valley Hall effect at all.

Response: we understand the point of the reviewer and we have considered other scenarios, for example we included a discussion where we compare different leading theories of the formation of the valley Hall effect: Berry curvature and spatially varying regions of broken sublattice symmetry and inhomogeneous strain. Unfortunately, we do not have experimental arguments to favor one of these.

I have some minor questions and comments:

(3) For the local measurement at low temperature (in Fig.2b), two resistance peaks at CNP are almost indistinguishable for 0° and 60° . (It is, in fact, hard for readers to identify since data are plotted in thick lines, hidden by

each other.) However, from the demonstrated band structure with relaxation in this manuscript (Fig.S11), the gap near zero energy in 60° is about twice larger than that of 0° . Do the authors see this difference in the experiment? From Fig. S5, the slopes of 0 and 60 degrees look similar. How to understand that?

Response: The thickness of the lines in the figure has been corrected and the overall aspect of the figure was modified to make more clear the differences at the CNP (insert) and satellite peaks. As you can now see in Fig.2d, the maximum of resistance at the CNP for 0° and 60° has at least a factor of six of difference. However, the energy gaps extracted by thermal activation and reported in Fig. 4b are the same between error bars. This result is very contrasting to our first numerical simulations where, as the reviewer mentions, the energy gap at 60° is about twice larger than at 0° . In the present version of the manuscript we used much more refined numerical simulations of the electronic band structure (fig. 4a) in which we include the intrinsic doping of the sample, obtained by the value of the energy gap at misalignment. This, as we explained in the last part of our manuscript leads to modifications of the band structure in which now the energy gap varies very little with angular alignment. The new simulations of the electronic band structure also present features that agree well with our experimental results.

(4) From the calculations, there is a gap at the satellite peak in the hole side. In the SI, the authors claim they didn't see a thermally activated gap in local transport measurement because the gap is small and "is washed out by disorder". But at the same time, the authors claim that the energy gaps of local and nonlocal at CNP "are at least are a factor of two larger than previously reported, this reaffirms the high quality of our samples". So can the authors estimate the disorder level? How does this disorder level compared with the devices in the cited references (the references in SI are missing, so I don't know which reference should I refer), and is it reasonable for this disorders wash out the gap of 5-10 meV at satellite peaks (I get this gap size from the calculated results in Fig. S11).

Response: We can compare the disorder between our samples, for this we have included Fig. S9 which compares the data of our samples. However, we are not able to compare with previously published data other than by the values of the extracted energy gap. In any case, the comparison here are qualitative and we do not have a reliable way to extract the real value of disorder to compare with the value of the energy gap. That's is why the fact that such a small gap is inaccessible because of disorder remains an hypotheses.

In our data we do not observe the energy gap at the hole side of the electronic band structure predicted by the numerical simulations. This might be because the predicted energy gap, which four times smaller than the one at the CNP, is out of reach with our current low temperature set-up. Experiments at lower temperatures, or with control of the applied displacement field, might allow to observe this energy gap. The reference problem have been solved.

(5) Since the nonlocal measurement is done at the insulating state, so the current pathway would be very complicated. I think the authors should be very careful. As the authors mentioned, they use "an operational amplifier to keep the voltage of our sample balanced and high impedance amplifiers to avoid any current leak". I believe the authors have done the test, but I encourage them to show the test results in the SI. For example, does the nonlocal resistance depends on the excitation current? Does the nonlocal signal agree with Onsager's relation?

Response: We have added a section in the SI where we clearly show that the non-local resistance do not depend on the current excitation, Fig S11.

(6) The authors mentioned that the localized edge states resulting from disorder give rise to the nonlocal signal at 30° , and the edge states disappear at 60° . That is very confusing that the edge states or the disorder can be different at different alignments with BN. Can the authors explain more about this? And a good experiment to test can be an edge length-dependent nonlocal measurement.

Response: We have added a short discussion about this in the manuscript: This behavior, unrelated to the valley Hall effect, could be explained either by the existence of localized and non-topological edge states resulting from edge disorder. Or by the presence of electronic jets separated of $\approx 60^\circ$ between them, and consequence of the trigonal warping of the bilayer graphene electronic band structure. The existence of these localized states contrasts with the ohmic response observed for the 60° of alignment. However, in both cases we can hypothesize that the periodic moiré potential may prevent the formation of the localized states (in the same way as artificial disorder generated by a scanning gate does) and that it will modify the trigonal warping of the electronic band structure.

(7) How can misaligned bilayer graphene on BN generate a gap, which is even larger than the aligned case? (FIG.2c).

Response: This can happen when some unintentional displacement field is created. We added a discussion about it in the main text. To summarize, our local charge transport measurements show an energy gap of $E_g^{30^\circ} \approx 7.5 \pm 1.5$ meV at 30° of alignment, compare to literature [Icking et al.,] this represents an unintentional displacement field of ~ 0.1 V/nm. This is not surprising since our devices do not have a top gate to screen external doping deposited on top of the device.

(7)b. Why the non-local signal exhibited at 30° is strongest? (FIG.2e).

Response: Our hypothesis (as we explain in the main text) is that at 30° we have either localized and non-topological edge states resulting from edge disorder which generates two parallel conduction mechanisms: i) an ohmic conduction and ii) localized states on the edges (non-topological) contributing to the charge transport. Or the presence of electronic jets separated of ≈ 60 degrees between them, and consequence of the trigonal warping of the bilayer graphene electronic band structure. In any of these cases, it is important to notice that the strongest signal is observed at 30° of alignment because the signal at 0° saturates (fully developed valley Hall effect), otherwise if it followed the intermedium temperatures increase rate it will surpass the signal of 30° at the lowest temperature.

(8). The authors should pay attention to the paper writing: (a). Please check the format of references in the supplementary thoroughly and eliminate those strange [?] typos.

Response: this has been corrected in the present version.

(b). In Fig.S5a and S5b, data are displayed on a logarithmic scale in very strange numerical labels, which are difficult for readers.

Response: This has been corrected now as well as the typos of the caption.

(c). As mentioned before, the curves in Fig.2b are too thick. This way of presenting data makes it difficult for readers.

Response: This has been corrected in the present version.

(d). In Fig. S8, the y-axis has two 0.1 labels.

Response: This has been corrected in the present version.

(e). In Fig.S5, it would be more comparable for the data if the y-axis can be unified into one physic quantity, but not σ for the local measurements and G for the non-local measurements. And please take care of the typos in the figure caption.

Response: This notation is only used in the case of the Arrhenius plot in the supplementary material and is used to be able to compare directly with previous literature.

REVIEWER #2 (REMARKS TO THE AUTHOR):

Arrighi *et al.*, reported on local and non-local electronic transport data obtained from the hBN/bilayer graphene/hBN Hall bar device, where the top hBN layer is controllably aligned at different angles with the underlying graphene. In such systems, the crystallographic alignment of two crystals with dissimilar lattice constant produces a periodic moire potential, which ultimately leads to the valley Hall effect associated with the gap opening and finite Berry curvature. In their study, the authors went one step forward and probed the $\pi/3$ rotational periodicity of the valley Hall effect assumed from the crystal symmetry. To do that, the relationship between the local and non-local signal at four different misalignment angles, 0, 30, 60 and 120 degrees, has been compared. Unexpectedly, at 60-degree misalignment, the non-local vs local signal dependence demonstrated the behaviour typical for ohmic transport (linear dependence) instead of the anticipated valley Hall effect observed for the complete alignment (cubic dependence).

The valley Hall effect was claimed to recover at 120 degrees. The authors attributed the absence of the valley Hall effect-like response at 60 degrees to the spatial inhomogeneity of the lattice strained in the moiré commensurate state. The potential presence of the inhomogeneity was illustrated by molecular dynamics simulation of the atomic relaxation of bilayer graphene/hBN moiré superlattice. However, the detailed explanation of how the existence of the commensurate state challenges the current explanation of the valley Hall effect and its relation to the Berry curvature and band gap is absent.

Response: Our claims of the observation of the valley Hall effect with a periodicity of hundred and twenty degrees of alignment challenges the current explanation of this by the existence of a Berry curvature. Our point is the fact that the energy gap is the same, between error bars, for all the aligned positions therefore the Berry curvature of them should be very similar. Unfortunately, besides our numerous effort we haven't found a theory that explains why the different alignments, and therefore different band structures allow for the formation or not of the valley Hall effect.

Another observed feature is the quadratic dependence of the non-local vs local signal for the fully misaligned case (30 degrees), untypical for ohmic behaviour expected for a fully suppressed valley Hall effect at this rotation angle. The authors attribute this to the possible co-existence of the disorder-induced non-topological edge states and bulk conduction but do not present a more quantitative explanation.

Response: by only using transport measurements is very difficult to be more quantitative in this. A good way to approach this will be to have local imaging of these currents, as the measurements of Aharon-Steinberg et al., combined with transport measurements on rotatable devices. We agree that this is a very interesting that should be investigated deeply but also believe that it is out of the scope of our manuscript.

Although the results reported in this paper are very interesting and must be conveyed to the scientific community, I do not recommend it for publication in its present form. The paper requires significant improvement in terms of the data quality and consistency of the suggested explanation. To proceed with the improvement, I would make the following suggestions on the content and interpretation of the reported data:

1) More RNL vs ρ data taken at different misalignment angles is needed. If the authors are correct in their interpretation based on the inequivalent atomic displacement for 0 and 60-degree rotation, then they should confirm the $2\pi/3$ periodicity for all three distinct dependencies of the non-local signal. Namely, data presented in Fig3b should be repeated for 0, 30, 60, 90, 120, 180, 210, 240, 270, 300 and 330 degrees. Note that the data shown in Fig.S7 does not clearly show that the cubic dependence of the non-local signal recovers at 120 degrees of rotation; thus, more data is needed to prove that.

Response: Unfortunately, our experimental setup is has been under intermittently repairs the last few months and do not allow us to make a full set of low temperature data. However, we have included room temperature data that shows clearly the hundred and twenty degrees periodicity of the local transport. These data, Fig 1c main text, demonstrate the 120 degrees periodicity on two main values, the resistance at the CNP and the gate voltage at which it can be observed Figs S4 and S5.

2) The trend of the RNL vs ρ data taken at 30 degrees should be explained self-consistently, considering the amount of disorder present in the sample. The authors emphasised the importance of disorder but did not attempt to quantify and compare it between the two samples. It should be assessed either through charge carrier mobility or the width of the Dirac peak at low temperature and correlated with the observed trends of the non-local signal.

Response: We added a comparison of the Dirac peak width for two of our samples, Figs. S9. However, there is not a straight forward way to compare this with the existence of non-topological localised edge states (previously observed by Aharon-Steinberg et al.).

3) A control experiment with no top hBN handle should be performed to understand the influence of the bottom hBN layer and eliminate the effect of the unintentional alignment of graphene with the bottom hBN. For consistency, the same samples should be used with the top handle dragged away from the active region of the Hall bar.

Response: Our control experiment is the 30° alignment. The lack of signals of alignment at this position of the handles let us know that the bottom BN is fully misaligned. This can be seen in Fig 2b and Fig. S3.

4) The suggested explanation should be re-evaluated considering the above experimental data.

Response: We have re-evaluated our experimental data in terms of the above listed points and finding a plausible explanation to our observations is still out of reach. We hope this work will encourage other experimental and theoretical works that will help to clarify the origin of the valley Hall effect in this system.

REVIEWER #3 (REMARKS TO THE AUTHOR):

The paper by Ribeiro-Palau and co-workers exploits a system with dynamically tunable twist angle to study bilayer graphene rotated with respect to h-BN. This technique was pioneered by her when she was at Columbia and is an important new experimental probe for these twisted 2D materials. The advantage of this technique is that it keeps all the sample properties unchanged isolating only the effect of the twist angle.

The present work compares bilayer graphene twisted at 0, 30 and 60 degrees relative to the top h-BN handle that can be changed in situ. These results as a methods paper would be interesting in and of itself even if the authors did not uncover any new physics.

With the new physics they find, the paper is potentially suitable for Nature Communications. The authors need to address some technical concerns to ensure that the conclusions are robust. But I am optimistic that the authors will be able to do this.

The new physics involves finding a valley-Hall effect appearing only for zero twist angle and not for either 30 degrees or 60 degrees. This is despite the three states having roughly the same energy gaps (within experimental error bars). The authors attribute this difference to differences in lattice relaxation, which is a plausible explanation. Without relaxation, the 0 degree and 60 degree states are “twins”; hence the title of the paper.

A: Experimental Claims

(1) At a very basic level, the non-local measurements are done at the charge neutrality point where disorder effects are strongest. It is not uncommon in this field to mark off a window close to CNP where disorder effects are dominant, and the behaviour is different from elsewhere. Can the authors be sure that their measurements are outside this “disorder window”?

Response: We added Fig. S16 to the SI, in this we plot the non-local versus local signals relation for different densities around the CNP. We noticed that although the saturation regime (fully developed valley Hall effect) changes the cubic trend remains the same. This means that the effect is robust and also that even outside of this disorder dominated window the behavior remains the same.

(2) While the local transport on the scale of Fig. 2b show identical behaviour for zero and sixty degrees, a closer look show that the local resistivity is very different, eg. in Fig. S5, the low temperature saturation at 5K is very different between the red and blue symbols. So is it really correct statement to say that this difference in relaxation only shows up in the non-local signal and not the local signal?

Response: The reviewer is right, it is not correct to say that the difference between 0° and 60° of alignment only show in the non-local. As we show in the main text and the SI, the differences appear even at room temperature, see Fig. 1c. However, these differences mean something only when comparing the measurements of different angles of the same sample, which means that by measuring the local transport on aligned samples of bilayer graphene/BN we won't be able to tell if the crystals are aligned at 0° or 60°. To discriminate between these two alignments only the combination of local and non-local measurements can give a straight answers about the alignment, through the observation of the valley Hall effect *i.e.*, the cubic relation between the non-local resistance and the local resistivity.

I agree that the 30-degree sample does not show a valley Hall effect (e.g. the black curve in Fig. 3b exceeds the theoretical maximum shown as a dashed line in Fig. 3a). And I accept that the authors have put forward only a speculative explanation for the $\rho^{1.6}$ scaling since this is not central to the main claims of the paper. However, the comparison with the 60-degree sample should be considered a bit more carefully. The main claim of this paper rests almost entirely on purple curve having $\sim \rho^1$ over the entire temperature range and the orange curve having three distinct regimes ($\sim \rho^2$, $\sim \rho^3$, and $\sim \rho^0$) here are some questions about this claim:

(3) Is the temperature range for the 3 curves in Fig. 3b the same?

Response: yes, as we mentioned in the caption of the figure “Both **a** and **b** are measurements between $T = 1.4$ K and $T = 110$ K.” Also the steps in temperature are the same.

(4) The authors write that the purple curve “can be adjusted by using only the geometry of our sample”. Was this experiment done with a different sample geometry? This would be a convincing argument.

Response: As mentioned before, unfortunately we were not able to perform complete sets of measurements at low temperatures. In any case changing the geometry of our samples, although very interesting, it is very complicated to do because of the limited area of the BN structure and graphite gate.

(5) Can the authors plot the 60-degree data also in Fig. 3a so that it can be compared with the zero-degree data on the same scale? In the larger scale of Fig. 3b, only the $\sim \rho^3$ regime can be clearly seen.

Response: both the 0° and 60° data are plotted in figure 3b. If we plot data from the 60° alignment in the Fig 3a we won't be able to show clearly the saturation regime, which corresponds to the fully developed valley Hall effect, or we will have to cut the data of 60° which won't be fair to compare both alignments. Therefore, we prefer to keep the figure as it is at the moment.

B: Theoretical Claims

The relaxation model is nice, but I hope the authors can do a bit more.

(1) What is it about the relaxation that keeps the local gaps the same (within error bars) between 0 and 60 but changes the non-local gap by more than a factor of 2.

Response: This is an excellent question to which we do not have a clear answer. This answer will be exactly the reason why we observe the valley Hall effect in only one alignment. Unfortunately, no theoretical model seems to explain this so far.

(2) The authors should make a qualitative argument why either having relatively more circular symmetry in the first layer or a larger atomic relaxation in the second layer for 0 degrees is more favourable for the valley Hall effect.

Response: In the new version of our manuscript we added a short discussion about the $2\pi/3$ symmetry of the stain in the second layer, this is compatible with the theory developed by Guinea et al., In this, the particular strain symmetry is able to produce a pseudo-magnetic field which is roughly homogeneous at the center of the moiré cell and able to separate the carriers depending on the valley they belong to. This could be the explanation we are looking for. However, only transport data do not allow us to proof this theory and remains an hypothesis. We hope our results inspire other experimental and theoretical works that will help to understand better this phenomena.

(3) Similarly, the authors should make a qualitative argument for what happens to the Berry curvature when there is relatively more circular symmetry in the first layer or a larger atomic relaxation in the second layer.

Response: As mentioned in the SI the Berry curvature also depends on the interlayer interaction, this means that the difference in atomic displacement between the layer can also play a role in the Berry curvature. However, the predominant term is the energy gap and without a strong variation of this one the changes on the Berry curvature are not really significant. For completion, we added numerical simulations of the Berry curvature to the SI, in this we cannot see any relevant difference between 0° and 60° .

C: Minor questions...

(1) Why are the satellite peaks washed away at room temperature? Can a simple thermal broadening model capture this effect, or is there something more going on?

Response: Simple thermal broadening can explain every well the absence of the satellite peaks at room temperature. We added two new figures to the supplementary material where we can see the temperature dependence for two samples, in these is clear that thermal broadening is dominating the response at room temperature. We have also

added a sentence to make this clear in the main text

(2) Most references in the supplemental materials appeared as [?]

Response: This issue has been solved in the present version.

REVIEWER COMMENTS

Reviewer #1 (Remarks to the Author):

The authors didn't answer two of my questions.

1. Reproduce the nonlocal measurement results in a second device;
2. Does the measurement follow the Onsager relation? In particular, if the authors reverse the current and voltage probes, are the nonlocal signals still the same?

These two questions are important because they will tell us if the measured signals are real or not. As a result, I cannot recommend publication.

Reviewer #2 (Remarks to the Author):

Dear Editor,

I am not completely satisfied with how the authors have addressed my comments. It is still difficult to understand why the authors could not find an alternative experimental setup for the low-temperature measurements outside their laboratory. An attempt to present the room temperature local resistance data as an additional argument in favour of the observed 120-degree periodicity of non-local (!) resistance poses more questions than answers and does not clarify the situation. I strongly recommend that the authors collaborate with other groups/colleagues with access to a working cryostat and confirm the 120-degree periodicity of the non-local resistance at low temperatures. After these low-temperature non-local data are included, the manuscript can be accepted for publication in Nature Communications. Sincerely,

Reviewer #3 (Remarks to the Author):

This is my second look at the manuscript by Arrighi et al. In the first round I was very favorable of the manuscript, but had some questions to confirm the picture presented by the authors.

Looking at the revised manuscript and the response to referees, I do not think these concerns have been completely addressed.

Most of the problem seems to be that the authors have had problems with their cryostat. Even with a working cryostat it would have been difficult to do all the checks proposed by the 3 Referees. Without

the working cryostat, these checks are impossible.

At this stage, one needs to make a judgement call. I think on the balance the paper should be published in Nature Communications. However, given the challenges in addressing referee concerns, the authors might consider toning down their claims leaving it open to alternate interpretations.

Just for the record, let me comment on which points in my earlier report the authors adequately addressed, and which ones still leave me with lingering doubts.

(E1) I asked whether the effect was robust away from the disorder-dominated regime close to charge neutrality. Note from Fig. S9, this regime is $3.3 \times 10^{10} \text{ cm}^{-2}$ for sample 1, and $1.2 \times 10^{11} \text{ cm}^{-2}$ for sample 2. From the capacitance estimates (below Fig. S2), this disorder window would be about 1V. Yet in Fig. S16, the authors take points at 0.01V, 0.02V, and 0.04V. It is not clear why the authors think such small deviations are “outside the disorder dominated window”.

(E2) Okay. The text was modified to address this concern.

(E3) Okay. The authors adequately addressed this.

(E4) The authors were unable to do this additional experiment. I appreciate this would be difficult even if their cryostat was working.

(E5) Even if the authors did not want to change the figure in the manuscript, they could have indulged this referee by making this plot in a way that could have been preserved in the referee correspondence that is often available alongside the manuscript. The purpose of such a figure would be to properly judge the claims made by the authors.

(T1) The authors were unable to comment on this question.

(T2) The authors provided a reasonable response to this question. Although I should mention that it is not very difficult to calculate the pseudomagnetic field from the atomic positions in Fig. 3a,b.

(T3) It seems that their extra work along this direction disfavored the Berry phase argument proposed in the original manuscript. I am happy with the way it is now discussed in the present manuscript.

Point-by-point response to the reviewers' comments Nat Comms

We thank the three reviewers for the second review of our manuscript and for the possibility to open the discussion again. We are very happy to let you know that our cryogenic system is back and that we have completed the data on sample IV for a total rotation of 180 deg (from -60 deg to 120 deg). We hope the reviewers will find answers to all their questions. Here a point by point answer:

REVIEWER #1:

We strongly apologize for not replying to this point sooner, we hope you can find all the answers here:

1. Reproduce the nonlocal measurement results in a second device;

Response: We have completed the data on Sample IV (Figure S20). In this you can find the local versus non-local data for the same device in a full rotation of 180 deg (from -60 deg to 120 deg). In Figure S20 you will see that the local and non-local signal have a cubic relation which can be recover every 120° of rotation.

2. Does the measurement follow the Onsager relation? In particular, if the authors reverse the current and voltage probes, are the nonlocal signals still the same?

Response: We have included figure S11, which demonstrates that not matter the measurement configuration the local versus non-local signal relation stays the same.

REVIEWER #2:

Response:We thank the reviewer for their encouragement as we wrote in the response to reviewer #1: We have completed the data on Sample IV (Figure S20). In this you can find the local versus non-local data for the same device in a full rotation of 180 deg (from -60 ° to 120 °). In Figure S20 you will see that the local and non-local signal have a cubic relation which can be recover every 120° of rotation.

REVIEWER #3 (REMARKS TO THE AUTHOR):

We reply point by point here:

(E1) I asked whether the effect was robust away from the disorder-dominated regime close to charge neutrality. Note from Fig. S9, this regime is $3.3 \times 10^{10} \text{ cm}^{-2}$ for sample 1, and $1.2 \times 10^{11} \text{ cm}^{-2}$ for sample 2. From the capacitance estimates (below Fig. S2), this disorder window would be about 1V. Yet in Fig. S16, the authors take points at 0.01V, 0.02V, and 0.04V. It is not clear why the authors think such small deviations are “outside the disorder dominated window”.

Response: We believe there is a misunderstanding which comes from a inconsistency of units from our side. The disorder-dominated regime close to the CNP is $\delta n = 3.3 \times 10^{10} \text{ cm}^{-2}$, which given the capacitive coupling of $6.4 \times 10^{11} \text{ V}^{-1} \text{ cm}^{-2}$ (previously expressed in $\text{V}^{-1} \text{ m}^{-2}$) gives a $\delta V_g = 0.052 \text{ V}$, which means that for $|n| > 1.65 \times 10^{10} \text{ cm}^{-2}$ we should be outside of this disorder region. We have now expressed the values of Fig.S17 in density to make it more clear.

(E4) The authors were unable to do this additional experiment. I appreciate this would be difficult even if their cryostat was working.

Response: As we wrote in the response to reviewer #1: We have completed the data on Sample IV (Figure S20). In this you can find the local versus non-local data for the same device in a full rotation of 180 deg (from -60 ° to 120 °). In Figure S20 you will see that the local and non-local signal have a cubic relation which can be recover every 120° of rotation.

(E5) Even if the authors did not want to change the figure in the manuscript, they could have indulged this referee by making this plot in a way that could have been preserved in the referee correspondence that is often available alongside the manuscript. The purpose of such a figure would be to properly judge the claims made by the authors.

Response: Here we provide the new plots requested by the reviewer.

FIG. 1. **Local and non-local signal sample I.** (Top) Local and non-local signal for sample I at 0° and 60° . (Bottom) zoom in the 0° data.

(T1) The authors were unable to comment on this question.

Question: (1) What is it about the relaxation that keeps the local gaps the same (within error bars) between 0 and 60 but changes the non-local gap by more than a factor of 2.

Response: Unfortunately, our current atomistic simulations do not allow us to get a deeper inside in the origin of the enhanced non-local gap (and therefore of the valley Hall state). We believe that answering this question will give the key to the understanding of the observation of the valley Hall effect in our considered system. We hence hope that our results encourage other theoretical and experimental works about this. In particular, new transport theories will be necessarily developed to properly take the details of atomic structure (i.e., atomic relaxation effects) into account.

REVIEWERS' COMMENTS

Reviewer #1 (Remarks to the Author):

The authors reproduced their main data in other devices at low temperatures. There are no technique issues. I agree with publication.

Reviewer #2 (Remarks to the Author):

Dear Editor,
I am satisfied with the correction made to the manuscript and it can now be published.
Sincerely,

Reviewer #3 (Remarks to the Author):

This is my third review of the manuscript by Arrighi and co-workers. In my two previous reviews I supported publication in Nature Communications, but highlighted several caveats. I am happy that the authors have managed to get their cryostat working again, and Fig. S11 and Fig. S20 address the reproducibility concerns raised by the other referees. I also very much appreciate Fig. S17 which was made to address my concerns.

I don't mean to nit-pick, but the data at $\pm 2.5 \times 10^{10} \text{ cm}^{-2}$ is still within Δn (which from Fig. S9 is $3.3 \times 10^{10} \text{ cm}^{-2}$). The authors could consider adding to Fig S17 one data set that is clearly outside the disorder window (even if this data does not show the same trend as the other curves).

What I like best about the current version is that it lays out the experimental evidence, discusses possibilities, but then leaves the door open for future theoretical and experimental work. There is a wealth of data in the supplemental material that will be extremely useful for future studies. And the main idea that lattice relaxation dramatically alters the symmetry between 0 and 60 degrees and presence of the valley Hall effect in only the first configuration is an important result. I am happy to support this manuscript for publication in Nature Communications.

Point-by-point response to the reviewers' comments Nat Comms

We thanks the three reviewers for their last review of our manuscript.

REVIEWER #3 (REMARKS TO THE AUTHOR):

We reply point the final point to address here:

“I don't mean to nit-pick, but the data at $\pm 2.5 \times 10^{10} \text{ cm}^{-2}$ is still within δn (which from Fig. S9 is $3.3 \times 10^{10} \text{ cm}^{-2}$). The authors could consider adding to Fig S17 one data set that is clearly outside the disorder window (even if this data does not show the same trend as the other curves)”

Response: Maybe there is still a small misunderstanding, the disorder-dominated regime close to the CNP is $\delta n = 3.3 \times 10^{10} \text{ cm}^{-2}$, which means that for $|n| > 1.65 \times 10^{10} \text{ cm}^{-2}$ we should be outside of this disorder region. We have modify supplementary figure 17 int he way shown below to make this point clear.

FIG. 1. **Cubic relation around the CNP for 0° , sample I.** The same cubic relation can be observed for values of densities close to the CNP. Insert: non-local measurement for 0° at 1.6 K, gray area represents the disorder dominated regime extracted in Supplementary Figure 9, the color of the dashed lines correspond to the different densities marked in the legend.